# Asciminib Mitigates DNA Damage Stress Signaling Induced by Cyclophosphamide in the Ovary

**DOI:** 10.3390/ijms22031395

**Published:** 2021-01-30

**Authors:** Luca Mattiello, Giulia Pucci, Francesco Marchetti, Marc Diederich, Stefania Gonfloni

**Affiliations:** 1Department of Biology, University of Rome Tor Vergata, via della Ricerca Scientifica, 00133 Rome, Italy; luca.mattiello@alice.it (L.M.); giulia.pucci94@gmail.com (G.P.); francescomarchetti@icloud.com (F.M.); 2College of Pharmacy, Seoul National University, 1 Gwanak-ro, Gwanak-gu, Seoul 08826, Korea; marcdiederich@snu.ac.kr

**Keywords:** ovarian reserve, cyclophosphamide, DNA damage response, drug repurposing, allosteric tyrosine kinase inhibitors, asciminib

## Abstract

Cancer treatments can often adversely affect the quality of life of young women. One of the most relevant negative impacts is the loss of fertility. Cyclophosphamide is one of the most detrimental chemotherapeutic drugs for the ovary. Cyclophosphamide may induce the destruction of dormant follicles while promoting follicle activation and growth. Herein, we demonstrate the in vivo protective effect of the allosteric Bcr-Abl tyrosine kinase inhibitor Asciminib on signaling pathways activated by cyclophosphamide in mouse ovaries. We also provide evidence that Asciminib does not interfere with the cytotoxic effect of cyclophosphamide in Michigan Cancer Foundation (MCF)7 breast cancer cells. Our data indicate that concomitant administration of Asciminib mitigates the cyclophosphamide-induced ovarian reserve loss without affecting the anticancer potential of cyclophosphamide. Taken together, these observations are relevant for the development of effective ferto-protective adjuvants to preserve the ovarian reserve from the damaging effects of cancer therapies.

## 1. Introduction

Chemotherapy, radiation, or combinations of them are commonly used in cancer therapy. Although they improve patient survival, cancer therapies can alter ovarian follicle growth and can damage the follicle reserve. Primordial follicle stockpiles are non-renewable and ovarian insufficiency and infertility are well-known side effects of cancer treatments [1,2,3]. The options that are currently available to preserve fertility rely upon the age of the patient and the urgency of cancer therapies. Preventing infertility through co-medication represents a pressing challenge concerning pre-pubertal patients with hematological cancers. To address such a challenge, we need to understand the molecular mechanisms underlying ovarian-reserve loss induced by cancer therapies. Alkylating agents are the most toxic to the ovaries, and Cyclophosphamide (Cy), which is routinely used against solid and hematological malignancies, is particularly toxic [4]. Recent studies have highlighted the primary target of Cy in vivo in murine ovaries [5,6,7,8]. In regard to possible causes of primordial-follicle loss induced by cyclophosphamide, two main hypotheses have been proposed. These hypotheses include accelerated follicle activation, (burn-out) [9], and a direct activation of the DNA damage response in the nuclei of reserve oocytes [6,7]. Extensive follicle activation, which is possibly due to a deregulation of pathways underlying the follicular quiescence, indicates the involvement of the phosphoinositol-3-kinase (PI3K)/protein kinase B (Akt)/forkhead box O3 (Foxo3a) signaling pathway. Additionally, cyclophosphamide exposure can induce the death of growing granulosa cells and the concomitant activation of the DNA damage response (DDR) in the nuclei of reserve oocytes. Compelling evidence indicates that DDR activation is concomitant with the presence of the Akt-phosphorylated Foxo3a in the nuclei of reserve oocytes [6]. These findings suggest that damaged primordial follicles also lose their quiescent status. It remains unknown if losing follicular dormancy is a prerequisite to trigger the DDR, and this mechanism requires further investigation. These observations support the hypothesis that damaging stress pathways are activated in a concomitant manner in both somatic and germ cells in response to chemotherapy. Indirect activation of primordial follicles due to a lack of suppressive factors such as anti-Müllerian hormone (AMH) can also be induced by degeneration of growing follicles [10,11,12]. As the physiological changes in the ovary reflect both direct and indirect effects of genotoxic assaults, the identification of the sentinel molecules directly involved in communicating stress signaling remains a daunting task. Despite this, understanding the molecular mechanisms underlying ovarian-reserve loss induced by cancer therapies remains essential for developing a more effective treatment to preserve the fertility of female patients. Inhibitors of the pathways involved in either follicle activation or the DDR should protect the ovarian reserve from the damaging effects of Cy. Recently, we evaluated the effect of transient administration of various DDR kinase inhibitors in regards to their ability to limit the Cy toxicity in vivo [6]. We determined that an allosteric c-Abl inhibitor (GNF2) was more effective than were ATP-competitive kinase inhibitors in protecting the ovarian reserve from Cy. Here, we investigated the effects of Asciminib (the first allosteric c-Abl inhibitor used in a Phase III trial) against Cy in vivo in a pre-pubertal mouse model.

## 2. Results

### 2.1. Asciminib Induced a Re-Localization of the C-Abl Tyrosine Kinase to the Perinuclear Zone

In this study, we evaluated the effects of Cy (or of an active metabolite 4-hydroperoxy-cyclophosphamide, 4-OH-Cy) alone and in combination with Asciminib, either in vivo in mice or in vitro using a model cell line for breast cancer (Michigan Cancer Foundation (MCF7)). First, we validated the re-localization of c-Abl after treatment with Asciminib, as it was previously described for the allosteric inhibitor GNF2 [13]. We monitored the c-Abl localization in a transgenic Mouse Embryonic Fibroblast (MEF, lacking the expression of c-Abl) line following transfection with a c-Abl expression vector. We evaluated the c-Abl localization using an immunofluorescence (IF) assay in transfected MEF cells that were treated with different c-Abl inhibitors. We tested either allosteric ligands, such as GNF2 and Asciminib, and one ATP-binding competitive inhibitor (imatinib). IF assays clearly revealed that Asciminib and GNF2 both induced a re-localization of c-Abl in the perinuclear zone, and this is indicated by the yellow arrows in Figure 1. In contrast, treatment with imatinib did not cause any enrichment of c-Abl kinase in the perinuclear zone.

### 2.2. Asciminib Modulated the DDR and the Follicle Activation Induced by Cy Exposure In Vivo

Next, we injected P7 mice with Cy alone (100 mg/kg) or in combination with increasing concentrations of Asciminib (0.1, 0.2, and 0.5 mg/kg, respectively). At different time points, the ovaries were dissected and analyzed using IF or Western blot (W.B.) assays. Co-treatment with Asciminib resulted in partial inhibition of TAp63 modification (commonly observed as a shift of TAp63 protein according to W.B. assay). In Figure 2A, we observed a partial prevention of TAp63 shift (see yellow arrows) 18 h after co-injection with Cy and Asciminib. We observed the phosphorylation of histone H2AX at Ser139 (γH2AX), an early marker of DDR in the ovarian lysates. Of note, co-treatment with Asciminib affected the γH2AX phosphorylation, as assessed by W.B. assay (Appendix A).

To assess if Asciminib affects DDR activation in primordial/primary oocytes, we monitored the phosphorylation of DDR sentinel proteins using IF assays performed on ovarian sections. We found that Asciminib attenuated DNA stress signaling induced by Cy in the ovarian reserve. Co-treated ovaries exhibited reduced staining for phospho-DNA-PK, γH2AX, and cleaved PARP in the nuclei of reserve oocytes (Figure 2B–D). Furthermore, co-treatment of reserve oocytes with Asciminib and Cy prevented nuclear AKT phosphorylation (Figure 2E). Taken together, these data demonstrate that Asciminib can affect both of these signaling pathways activated by Cy in the ovary.

### 2.3. Asciminib Protected the Ovarian Reserve Following Cy Treatment

Next, we injected female P6 pups with Cy (100 mg/kg) alone or in combination with Asciminib (0.25 mg/kg). Three days after injection, we collected ovaries to perform immunohistochemistry (IHC) assays using an antibody against the cytoplasmic germ cell antigen (Msy2) (red) Figure 3. We counted the follicle reserve from mid-ovary sections of different ovaries. IHC assays examining ovarian sections revealed a massive depletion of primordial and primary follicles in Cy-treated mice, and Cy + Asciminib co-treatment significantly rescued reserve follicles. Follicle protection was dependent upon the concentration of Asciminib that was administrated (Appendix A). A higher dosage of Asciminib (1 mg/kg) did not prevent follicle death induced by Cy, and appeared to exert a minor toxic effect.

### 2.4. Asciminib Did Not Prevent the DNA Damage Induced by Cy in MCF7

A clinically used ferto-protective drug should not interfere with the therapeutic effect of DNA-damaging chemotherapies. We validated this assumption by assessing the effect of Asciminib on 4-hydroperoxy-cyclophosphamide (4-OH-Cy)-treated MCF7 breast tumor cells. Our results demonstrate that the co-treatment with Asciminib did not affect 4-OH-Cy-induced phosphorylation of DDR marker proteins like ATM, γH2AX, or p53 (Figure 4A). Additionally, single-cell gel electrophoresis (Comet) assays revealed that Asciminib did not interfere with the DNA-damaging effect of 4-OH-Cy (Figure 4B). Finally, the co-administration of Asciminib did not affect the cytotoxic effect of 4-OH-Cy (Figure 4C). Taken together, these data support the potential use of Asciminib as a *ferto-protective* drug without abrogating the cytotoxic effect of 4-OH-CY.

## 3. Discussion

In young women, chemotherapy regimens increase the risk of premature ovarian failure (POF) and infertility. Genotoxic agents exert two different effects on ovarian function, where the first effect is immediate and induces amenorrhea and the loss of growing follicles, and the second effect is long-term and induces a loss of the ovarian reserve. While the first effect is reversible, the loss of the follicle reserve is permanent, and leads to infertility. Premature depletion of the ovarian reserve following cancer therapies has been recently reviewed [2,14,15,16,17]. In recent years, animal models have helped to define the cellular and molecular mechanisms underlying chemotherapy cytotoxicity [18,19,20,21,22,23]. Small molecules were tested in vivo in combination with the purpose of limiting the damaging effects of chemotherapeutic agents [5,7,24,25,26]. Such experiments were designed with the purpose of developing effective pharmacological options to prevent follicle loss at the time of treatment. The outcome of such efforts may be useful for identifying small molecules (ferto-protective) that would provide several advantages for fertility conservation techniques. Ferto-protective drugs [27] can be suitable for patients of all ages, and avoid the use of invasive hormonal and surgical procedures. Notably, such small molecules should not interfere with chemotherapeutic treatments while preventing the endocrine side effects of premature ovarian failure and infertility [9].

Primordial follicles maintain their genomic integrity for several decades in humans without relying on the classical DNA quality check controls that are present in somatic cells and occur during the cell cycle [28]. Recent findings from Rinaldi et al. suggest that the oocytes, upon reaching a threshold level of unrepaired double strand breaks (DSBs) after irradiation, can be eliminated through signaling pathways that require both p53 and TAp63α transcription factors [29]. Recently, we performed ChIP experiments that revealed that p53 is primarily involved in the apoptotic response induced by Cy in vivo in the ovary [6]. Agents used to protect the ovaries should interfere with the pathways activated by chemotherapy. The non-receptor tyrosine kinase c-Abl is expressed within the ovary [30]. Although its role in gonads remains to be identified, in cells, c-Abl regulates diverse cellular processes including responses to DNA damage and oxidative stress [31,32]. Compelling evidence suggests a role for c-Abl in the oocyte degeneration induced by chemotherapy [24,33,34]. In mice, pharmacological inhibition of c-Abl counteracted the cytotoxic effect of cisplatin [24,25,33] and cyclophosphamide [6]. Recently, we compared the effect of transient administration of various inhibitors against apical kinases involved in DDR [6]. An allosteric c-Abl inhibitor (GNF2) was more effective than were ATP-competitive kinase inhibitors in protecting the primordial follicle reserve from Cy [6]. In contrast to ATP-competitive kinase inhibitors, allosteric c-Abl inhibitors do not target multiple kinases, thus limiting any off-target activity (recently reviewed by [35]). Allosteric inhibitors bind to a distal pocket in the large lobe of the c-Abl catalytic domain, which is far from the ATP-binding site. Of note, small allosteric ligands can induce conformational changes even in the ATP binding pocket of the kinase, thus preventing its active conformation [36]. Understanding the mechanism of action of GNF2 allowed for the discovery of new myristate-binding pocket compounds and their optimization as potential drugs [37,38]. These studies culminated in the discovery of Asciminib as a candidate for clinical use [39].

Based on these previous findings, in this study, we tested Asciminib against the damaging effects of cyclophosphamide. Asciminib is a more selective inhibitor compared with GNF2 [40]. Importantly, this inhibitor has already been used in Phase III assays (NCT03106779) against resistant forms of chronic myeloid leukemia (CML) [41]. Here, we revealed that the co-administration of Asciminib exerted a protective effect on the ovarian reserve, as assessed by immunohistochemistry (IHC) performed three days after Cy injection. We also determined that Asciminib does not counteract the genotoxic effect exerted by a Cy active metabolite (4-OH-Cy) on the MCF7 breast cancer cell line. Taken together, these data support the potential use of Asciminib as a ferto-adjuvant during chemotherapeutic regimens.

## 4. Materials and Methods

### 4.1. Animals and Injection

In accordance with the ethical standards of the Declaration of Helsinki, in compliance with our institutional animal-care guidelines, and following national and international directives (Italian Legislative Decree 26/2014, Directive 2010/62/E.U. of the European Parliament and of the Council), the experiments involving mice and their care were conducted at the Interdepartmental Service Centre-Station for Animal Technology (STA, Rome, Italy), University of Rome “Tor Vergata”. This trial was approved by Ministero della Salute, Direzione Generale della Sanità Animale e dei Farmaci Veterinari Ufficio VI (ex-DGSA-benessere animale) (23 September 2015, duration: 36 months) and was registered by Ministero della Salute DGFA 0024410-P-, 25 September 2015. Project identification code: 1007/2015-PR). We collected ovaries from newborn CD-1 mice (Charles River) at 6–8 days of age. We treated mice by intraperitoneal (I.P.) injection of PBS or Cy (100 mg per kg of body weight). Mice were pre-treated with Asciminib (0.1–0.5 mg per kg of body weight) prior to Cy injection. Cy (BAXTER) was freshly prepared at 40 mg/mL in PBS. We dissolved Asciminib in DMSO.

### 4.2. Immunohistochemistry, Follicle Counting, and Statistical Analysis

We prepared sections from ovaries that were fixed in MetaCarnoy solution, embedded in paraffin, and cut in slices of 7 μm in thickness. Sections were then dewaxed, rehydrated, and microwaved. PBS/Triton 0.2% permeabilization was performed prior to incubation with the MSY-2 antibody (Santa Cruz Biotechnology, Inc. San Diego, CA, USA). We used the Immunocruz staining system (Santa Cruz, sc-2023) and 3-aminoethyl-9-ethylcarbazole as a substrate (AEC, Sigma-Aldrich, Saint Louis, MO, USA) as previously described [6]. Quantitative analysis of primordial and primary or secondary follicles was based on histological analysis and counting of Msy2-positive germ cells in mid-ovary sections. For the counting of each ovary, only several central slices were used (10–15). The quantification of primordial/primary follicle reserve is reported as the mean of the immature follicles (primordial plus primary follicles) per individual ovary. Each point on the scatter plot represents the average values for each ovary (Mean value ± S.D.). One-way (ANOVA), with Turkey’s multiple comparison or unpaired Student’s t-tests, according to the data characteristics, was performed using GraphPad PRISM 6 (GraphPad Software Inc. San Diego, CA, USA) (* *p* < 0.05; ** *p* < 0.01; *** *p* < 0.001).

### 4.3. Immunofluorescence and Immunoblot Analysis

Ovaries were fixed in MetaCarnoy solution, paraffin-embedded, and cut into slices of 7 μm in thickness. Dewaxing, re-hydration, antigen unmasking and immunofluorescence were performed as previously described [6]. We homogenized P7 dry-ice-frozen ovaries using a mini-pestle in ice-cold lysis buffer, as previously described [6]. We loaded equal aliquots of protein extract (equivalent of one to three gonads) onto 6%, 8%, or 12% SDS-PAGE gels, and, following electrophoresis, we transferred the proteins to a nitrocellulose membrane (Amersham BioScience, UK Ldt, Buckinghamshire, United Kingdom,). Immunoblot densitometry was performed using ImageJ software (NHI, National Institute of Health, Bethesda, MD, USA).

### 4.4. Cell Culture

MEF Abl−/− cells were kindly gifted to us by Anthony Koleske’s lab (Yale School of Medicine, CT, USA). MCF7 cells were kindly provided by the Barilà group (IRCCS—Fondazione Santa Lucia, Rome, Italy). Cells were grown in a DMEM medium (GIBCO) supplemented with penicillin/streptomycin (100 U/mL Lonza) and 15% FBS (Mef Abl−/−) or 10% FBS (for MCF7 cells). Cells were preserved in a controlled humidified atmosphere containing 5% CO_2_ at 37 °C.

### 4.5. Cell Transfection and Immunofluorescence Assay

MEF Abl−/− cells were grown in a six-wells plate on cover slips, and transfected with a plasmid encoding the wild-type c-Abl 1b isoform for 16 h (at low passages) using Lipofectamine 2000 DNA Transfection Reagent (Invitrogen, Waltham, MA, USA), according to the manufacturer’s instructions. During this time, the cells were also treated with small-molecule c-Abl compounds alone (GNF-2, imatinib, and Asciminib) for 4 h and then analyzed by immunofluorescence.

MCF7 cells cultured in a six-wells plate on cover slips were incubated for 60 min with the allosteric c-Abl inhibitors, Asciminib (0.5 μM), or GNF-2 (10 μM), and then treated with 4-hydroperoxycyclophosphamide (4-OH-Cy) (sc-206885) for 4 h prior to subsequent immunofluorescence analysis.

Cells were grown on coverslips, then washed in PBS and fixed with 4% paraformaldehyde for 10 min at room temperature. Cells were incubated with a solution containing Triton X-100 (0.5%), blocked for 2 h with a blocking solution (PBS, Triton X-100 0.1%, BSA 5%), and then incubated with primary antibodies against p-ATM, γH2AX, and p-p53 for 60 min. Incubation (30 min) was performed with Alexa 488-goat anti-rabbit (Invitrogen) and Alexa 555-goat anti-mouse (Life Technologies, Monza, Italy) antibodies followed by 1 μg/mL 4,6-diamidino-2-phenylindole dihydrochloride (Thermo Fischer Scientific, Rodano, Milano, Italy) for the staining of the nuclei. Fluorescence images were obtained using a Leica DMR Fluorescence Microscope (Leica Microsystems GmbH, Wetzlar, Germany). Perinuclear fluorescence analysis was performed using the FIJI software. The particular regions of interest (ROIs) were identified using a DAPI signal to locate the edge of the nuclear region. From this edge, a 30-pixel-wide area was isolated using the fixed ellipse command, and then the ROI was used to measure the mean fluorescence intensity.

### 4.6. Single-Cell Gel Electrophoresis Assay (Comet Assay)

Breast cancer cells (MCF7) cultured in 60 mm dishes were incubated for 60 min with the allosteric inhibitors Asciminib (0.5 μM) or GNF-2 (10 μM), and treated with 4-hydroperoxycyclophosphamide (OH-Cy) for 4 h. To prevent the occurrence of additional DNA damage, all subsequent procedures were conducted under dim light, as previously described [42]. The cells were washed with PBS twice, and 20 μL of each cellular lysate was mixed with 730 μL of 0.5% low-melting agarose solution. One tenth of this volume was added drop-by-drop to slides coated with 1% normal melting agarose. Cover-slipped slides were placed on ice for 5 min, and the coverslips were then gently eliminated. We followed all of the steps as previously described in [42]. After washing with deionized water, the slides were transferred into an electrophoresis chamber filled with alkaline electrophoretic solution [42]. Electrophoresis was performed in the same solution [42]. The slides were then washed twice with a neutralization buffer (Tris-HCl 0.4 M pH 7.5), fixed in 70% ethanol for 10 min, and dried at room temperature. Staining was applied immediately prior to microscopic analysis, and the slides were stained with GelRed (Sigma-Alrich Saint Louis, MO, USA) in the dark. Images of the comets were analyzed using Comet Score software; approximately 300 cells per slide were counted. DNA fragmentation was evaluated according to the percentage of DNA in the tail of the comet (% DNA tail).

### 4.7. MTS Assay

MCF7 cells were seeded into 96-wells plates at a density of 10,000 cells per well (200 µL), and treated with the following: vehicle control (DMSO) or OH-Cy, alone or in combination with Asciminib (0.5 μM). The cells were treated for 24 h. For the MTS assay, the Promega CellTiter 96^®^ AQueous One Solution Cell Proliferation Assay was used following the manufacturer’s instructions. The absorbance of cells for each condition was detected at 490 nM using a microplate reader (Sunrise Tecan microplate reader).

### 4.8. Reagents

We purchased antibodies from Santa Cruz (Msy-2 sc-21316); p-AKT (T308) sc-16646-R)), Millipore (p-H2AX γH2AX, 05–636), H2AX 07–627), SIGMA (p-DNA-PK (S2056) SAB4504169), Rockland (p-ATM (S1981), 200–301–500), and Cell Signaling Technology (antibody for p-P53 (S15) (9284) and cleaved PARP (9544). Polyclonal antibody for p63 was created in our laboratory using rabbit serum. Secondary antibodies for immunoblotting analysis were purchased from Jackson ImmunoResearch. All antibodies were diluted in a blocking solution according to the specific assay protocol that included 5% BSA in PBS Tween 0.05% for immunoblot analysis and 1% glycine, 5% FBS, and 5% NGS for immunofluorescence assays.

## Figures and Tables

**Figure 1 ijms-22-01395-f001:**
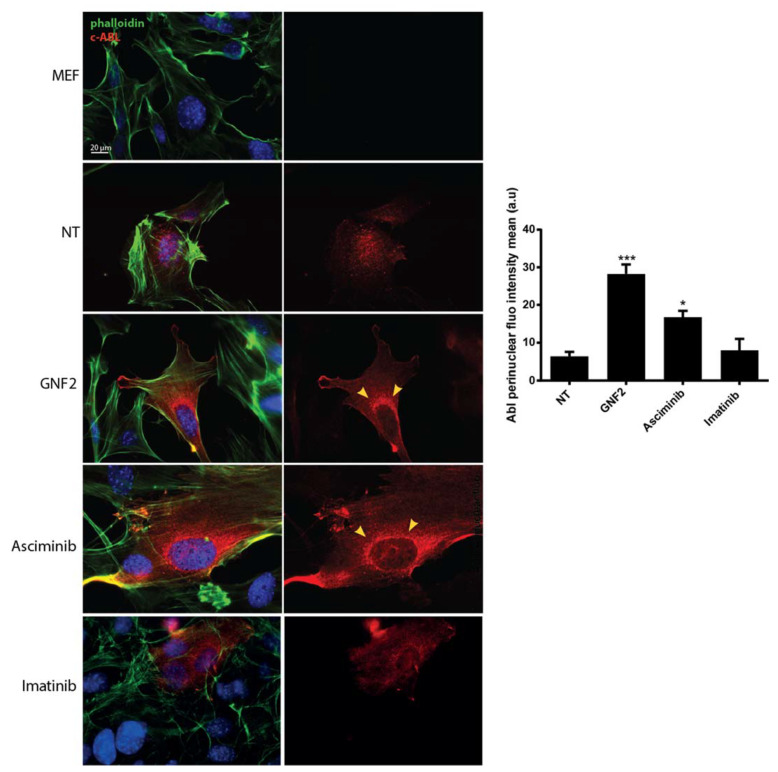
Allosteric inhibitors caused an enrichment of c-Abl tyrosine kinase in the perinuclear zone. Immunofluorescence (IF) assay analyzing Mouse Embryonic Fibroblast (MEF) c-Abl−/− cells transiently transfected with an expression vector for c-Abl. The yellow arrows indicate the re-localization of c-Abl tyrosine kinase in the perinuclear zone that was induced either by Asciminib or GNF2 exposure. The bar column represents mean ± s.e.m. Statistical analysis were performed using one-way analysis of variance (ANOVA); GraphPad PRISM 6 (GraphPad Software Inc. San Diego, CA, USA) * *p* < 0.05, *** *p* < 0.001 compared to the MEF transfected and not treated (NT) group. Scale bar 20 μm.

**Figure 2 ijms-22-01395-f002:**
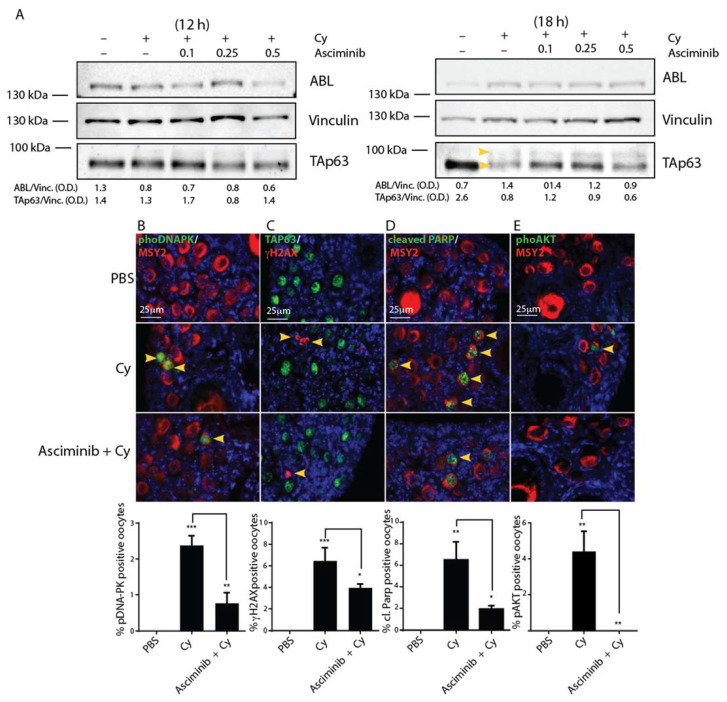
Asciminib mitigated both the DNA damage response (DDR) and apoptosis induced by Cyclophosphamide Cy in the ovarian reserve. (**A**) Western blot (W.B.) analysis of ovarian lysates from female pups injected with Cy 100 mg/kg, alone or in combination with various doses of Asciminib, collected at different time points. P7 mice were injected with vehicle (PBS) or Cy (100 mg/kg) alone or in tandem with Asciminib (0.25 mg/kg) and sacrificed within 12 h and 18 h after injection. Numerical values, under the blots, represent band densitometries that were normalized to the housekeeping gene. TAp63 shift is indicated by yellow arrows on the W.B. (**B**) DNA-PK activation was evaluated by IF assay using phospho-specific antibodies (indicated by yellow arrows) (green), and Msy2 (red) was used as a cytoplasmic marker for germ cells. (**C**) An IF assay, using phospho-specific antibodies (red), was performed to investigate γH2AX phosphorylation (indicated by yellow arrows), and we used p63 (green) as a nuclear marker for germ cells. (**D**) Follicle reserve apoptosis was evaluated by IF assay using antibodies against cleaved PARP (green) (indicated by yellow arrows), and we used Msy2 (red) as a cytoplasmic marker for germ cells. (**E**) Follicular activation was assessed by IF assay with specific phospho-antibodies for AKT (T308) (green) (indicated by one yellow arrow) and Msy2 (red). Quantitative assessment was performed on middle ovarian sections (6–8) derived from distinct ovaries. Scale Bar magnification, 25 μm. The bar column represents mean ± s.d. Statistical significance (* *p* < 0.05; ** *p* < 0.01; *** *p* < 0.001), as obtained by one-way analysis of variance (ANOVA), is shown in comparison to the PBS-treated group.

**Figure 3 ijms-22-01395-f003:**
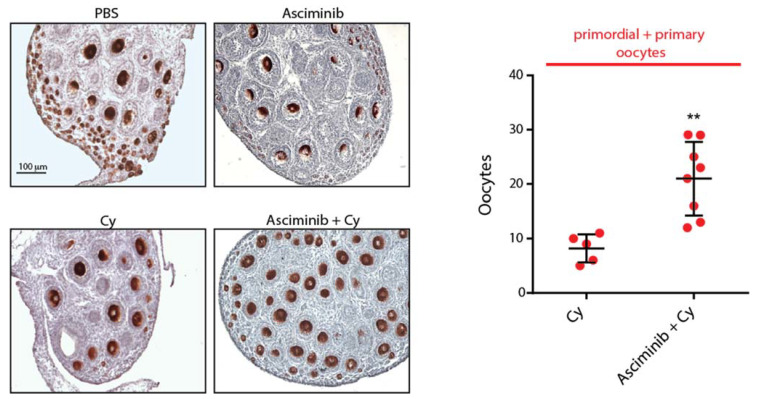
Asciminib protected the ovarian reserve from Cy treatment. Ovaries from each experimental group were dissected three days after injection (mice were injected with Cy 100 mg/kg alone or in tandem with Asciminib 0.25 mg/kg) and then analyzed by immunohistochemistry (IHC) assays using Msy2 antibodies. Independent experiments were performed, and several ovaries were analyzed. In the box plot, each dot represents the average primordial + primary follicle numbers per section of each gonad analyzed. One-way analysis of variance (ANOVA) was performed to determine statistical significance (** *p* < 0.01 compared to Cy 100 mg/kg). Scale Bar, 100 μm.

**Figure 4 ijms-22-01395-f004:**
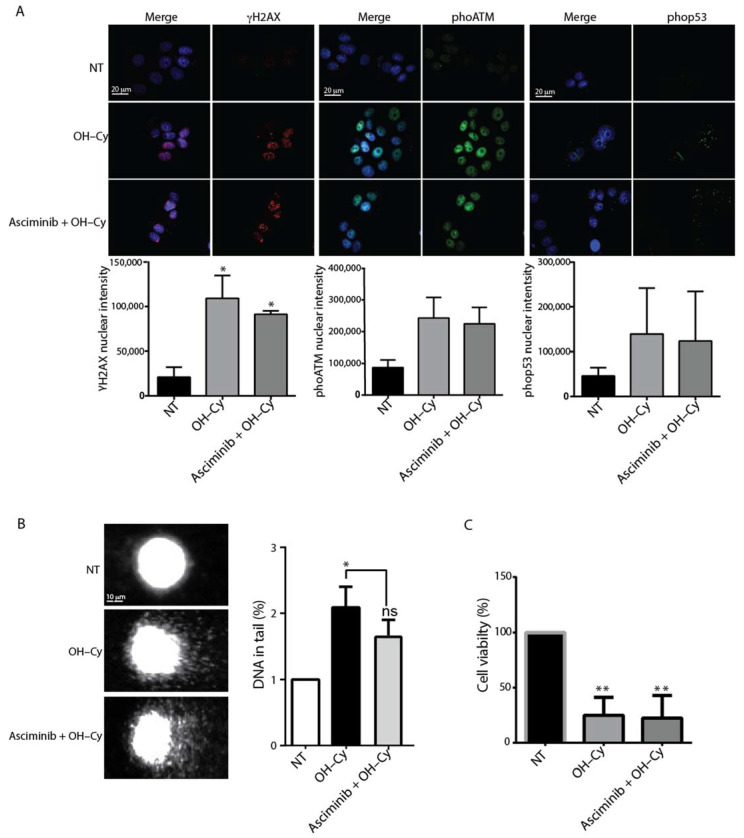
Asciminib did not prevent the cytotoxicity of 4-OH-Cy in Michigan Cancer Foundation (MCF)7. MCF7 cells were initially treated with Asciminib (0.5 μm). After 1 h, 4-OH-Cy was added in the medium (4-OH-Cy 10 μM for comet and IF assay and 4-OH-Cy 50 μM for MTS assay. (**A**) DDR signaling was evaluated by IF after 4 h following 4-OH-Cy treatment using specific phospho-antibodies for sentinel proteins. Bar columns represent mean ± s.e.m., and nuclear fluorescence intensity was evaluated using ImageJ software. Scale bar 20 μm. (**B**) DNA fragmentation was assessed by comet assay following 4 h of 4-OH-Cy treatment, and the DNA percentage according to tail quantification was evaluated by Comet Score software. Bar columns represent mean ± s.e.m.; Scale bar 10 μm. (**C**) Drug toxicity was measured according to MTS assay after 48 h of 4-OH-Cy treatment. Bar columns represent mean ± s.e.m (* *p* < 0.05; ** *p* < 0.01 compared to Not Treated cells, ns > 0.5).

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
