# Peer review of "Asciminib Mitigates DNA Damage Stress Signaling Induced by Cyclophosphamide in the Ovary"

_ijms, 2021, doi:10.3390/ijms22031395_

Round 1

Reviewer 1 Report

The authors have addressed all my comments and significantly improved the manuscript. The introduction and discussion parts are now well written. The authors also added more details in the experimental methods. I do not have any more queries. The manuscript is now acceptable. 

Reviewer 2 Report

The authors have responded to reviewers' queries.

In the introduction, the authors note "alter the follicle growth".  Could they change to ovarian follicle growth? 

This manuscript is a resubmission of an earlier submission. The following is a list of the peer review reports and author responses from that submission.

Round 1

Reviewer 1 Report

In the study by Mattiello et al, the authors reported ferto-protective adjuvant effects of asciminib during anti-cancer treatment with cyclophosphamide (Cy). Cy is a clinically approved drug for the treatment of breast and ovarian cancer amongst other indications that often have serious side effects of infertility in young women. The authors showed that co-administration of asciminib and Cy reverts the adverse effect to some extent without preventing the anticancer potential of Cy.

While the main claim of this paper is very encouraging, promising and important, there are some major concerns that must be addressed. A major revision is recommended before the manuscript is considered for publication. Specific comments are listed below:

  • The introduction and discussion sections are very brief. There was not adequate background information on the several pathways/signaling proteins that later came into the experimental sections. The authors also failed to cite and acknowledge relevant work in the same field.
  • It is surprising that most of the adverse effects of Cy were reported for acute (short term) treatment cases. In reality, infertility does not occur that fast and involves a series of other events. The authors should clarify why 18hrs or 3days treatment regimens were selected.
  • Fig 1. No ‘yellow arrows’ were shown on the figure as mentioned in the legends. Figure column should indicate which rows are WT or transfected with c-Abl. The graph should indicate statistical analysis (p value).
  • Fig 2. A shift in molecular weight as an indication of phosphorylation is not a definitive test. The authors should use phosphor-specific antibody.
  • Fig 3. Primordial cells often remain quiescent compared to primary follicular cells. Thus, these two types of follicle may not respond equality to anticancer drugs. The authors should report quantification of these two categories separately, and not represent as a combined number.
  • It was not clear how the authors came up with the idea that the combination with asciminib only would reverse the adverse effect of Cy. Was it a result of drug combination screening? The authors should describe the rationale in the discussion.
  • What is the anti-cancer (DDR, etc.) effects of Cy, asciminib, and the combo on healthy/normal ovarian follicular/fibroblast cells? The authors reported the same for MCF7 only.

Reviewer 2 Report

The authors of this original article provide a possible strategy to counteract the infertility caused by the anticancer drug cyclophosphamide (Cy) in women, trying to better understand the molecular mechanisms able to impair the signaling pathways involved in the death of follicole reserve and damage of both somatic and germ cells.

The authors have already focused on this field of studies in recent years and herein they have conducted experimentsin vivo and in vitro, supported by immunofluorescence and western blot assays, to evidence the role of the allosteric inhibitor of BCR-Abl kinase Asciminib combined with Cy in the ovary. Indeed, this molecule seems to be able to mitigate the destructive potential of the Cy on ovarian reserve acting on the signaling pathway triggered by Cy so as to prevent the loss of fertility in women. Interestingly, the same chemotherapy-mediated depletion has not been observed in cancer breast cellsin vitro, so Asciminib could be a response to the urgency to preserve the fertility of female patients.

As minor revision: There are several typos and some abbreviations were missed both in abstract and in the text, a through proof-reading and editing would be useful as well as improving the quality of the figures.

Reviewer 3 Report

The purpose of this manuscript was to "demonstrate the invivo protective effect of the allosteric Bcr-Abl tyrosine kinase inhibitor Asciminib on signaling pathways activated by cyclophosphamide in mouse ovaries."  This was a prospective, controlled trial using a murine model and invitro cell culture.

1.  Did the authors measure 4-hydroperoxy-cyclophosphamide in the mice treated with cyclophosphamide and the varying doses of Asciminib?  Could Asciminib interfere with formation of the active metabolite of cyclophosphamide?

2. In the introduction, please write the purpose of this manuscript is to .....

3.   In figure 1, I had a difficult time seeing the yellow arrows?  Please use different color, make larger or enhance contrast?

4.  In the Materials and Methods section could the authors please supply a figure of an algorithm of the treatment of the various groups of mice, and when and which tests were performed?

5. Could the authors please supply a table listing the various antibodies used (like MSY-2 antibody) and what these antibodies detect?  Like use of Msy2 as a cytoplasmic marker for germ cells.